# Advances in Transdermal Drug Delivery Systems and Clinical Applications in Inflammatory Skin Diseases

**DOI:** 10.3390/pharmaceutics17060746

**Published:** 2025-06-06

**Authors:** Sizhuo Liu, Tinghan Deng, Hongbin Cheng, Jun Lu, Jingping Wu

**Affiliations:** 1School of Clinical Medicine, Chengdu University of Traditional Chinese Medicine, Chengdu 610032, China; liusique@stu.cdutcm.edu.cn (S.L.); 201650501034@stu.cdutcm.edu.cn (T.D.); 2Traditional Chinese Medicine Hospital of Sichuan Province, Chengdu 610072, China; chenghongbin@cdutcm.edu.cn; 3College of Pharmacy, Chengdu University of Traditional Chinese Medicine, Chengdu 610032, China

**Keywords:** transdermal drug delivery system, inflammation, skin, nanocarrier, atopic dermatitis, psoriasis, acne

## Abstract

Inflammatory skin diseases are highly prevalent conditions characterized by complex immune responses that result in skin tissue damage and pain, significantly impacting patients’ physical health. Traditional therapeutic approaches, including oral administration and injections, continue to exhibit inherent limitations. Consequently, there is growing interest in exploring alternative drug delivery systems that offer more effective, targeted, and patient-friendly therapeutic options. Transdermal administration emerges as a promising solution for managing inflammatory skin diseases, facilitating sustained drug release, and reducing the frequency of dosing. This review provides a comprehensive overview of the skin barrier and critically summarizes clinically adopted transdermal drug delivery systems (TDDSs), including sonophoresis, iontophoresis, chemical penetration enhancers, and electroporation. Particular emphasis is placed on emerging advances in microneedle- and nanocarrier-facilitated transdermal delivery strategies. Moreover, the article synthesizes recent fundamental evidence regarding the application of TDDSs in the treatment of atopic dermatitis, psoriasis, and acne. This review examines fundamental research evaluating various transdermal drug delivery systems for the treatment of major inflammatory skin diseases, with an emphasis on their mechanisms of action, advantages, challenges, and future directions. Transdermal drug delivery systems hold the potential to deliver more efficient and safer treatment and management strategies for patients afflicted with inflammatory skin diseases.

## 1. Introduction

Over the past few decades, drug delivery systems have undergone rapid advancements [1]. Among the diverse strategies developed to enhance treatment outcomes and patient compliance, the transdermal drug delivery system (TDDS) has garnered widespread attention [2,3]. These systems, which utilize transdermal administration, offer substantial advantages over conventional oral or injectable formulations, including improved bioavailability, controlled drug release, and a non-invasive delivery method [4,5]. One of the most promising applications of transdermal drug delivery lies in the management of inflammatory skin diseases—chronic and often debilitating conditions that affect millions of individuals worldwide.

Inflammatory skin diseases, such as atopic dermatitis, psoriasis, and acne, are characterized by complex immune responses that lead to skin tissue damage, pain, and functional impairment [6,7,8,9]. These conditions not only compromise patients’ physical health but also exert a profound impact on their quality of life and contribute to escalating healthcare costs [10]. Traditional therapeutic approaches for inflammatory skin diseases predominantly encompass topical and systemic therapies. Topical therapies typically involve the direct application of medications to the affected skin lesions. While relatively safe, these treatments often leave the skin feeling greasy, resulting in reduced patient compliance. Moreover, drugs applied directly to the epidermis face significant challenges in penetrating the skin barrier, necessitating frequent application and prolonged onset times, which many patients find difficult to sustain. The effort and cost associated with managing skin conditions through topical therapies are particularly burdensome for individuals with severe inflammatory skin diseases or those seeking rapid symptom relief [11,12]. Systemic therapies, often administered orally or via injection, present an alternative approach. However, as these treatments do not target the affected area directly, they may give rise to additional health concerns. For instance, the nephrotoxicity of systemic drugs mandates regular monitoring of blood and liver function. Certain systemic treatments, such as cyclosporine, used to suppress the immune system and slow skin cell growth in psoriasis management, pose challenges for patients with compromised immune systems—such as lactating women—and can lead to side effects, including kidney dysfunction, hypertension, nausea, and headaches. Similarly, medications like hydroxyurea, originally developed for cancer treatment, may induce bone marrow suppression, reduced red blood cell counts, and an elevated risk of skin cancer when repurposed for inflammatory skin diseases [13]. Consequently, there is increasing interest in exploring alternative drug delivery systems capable of providing more effective, targeted, and patient-friendly treatments.

Transdermal drug delivery offers a promising solution to these challenges. Effective therapy requires delivering medications to the subcutaneous lesion site at the appropriate concentration and dosage [14,15]. Transdermal drug delivery targets drugs to the deeper dermis and subcutaneous layers, leveraging the skin as a delivery route to bypass the gastrointestinal tract and first-pass metabolism in the liver, thereby enhancing drug bioavailability. Furthermore, these systems enable sustained drug release over extended periods, reducing the need for frequent dosing and minimizing the risk of side effects. Given that the skin is an accessible and non-invasive organ, transdermal drug delivery represents an attractive option for patients, particularly those with chronic conditions necessitating long-term management [16,17].

The evolution of transdermal systems has been propelled by advancements in materials science, pharmacokinetics, and biotechnology. Technologies such as passive diffusion, iontophoresis, microneedles, and nanoparticles have been integrated into transdermal patches and devices to enhance drug penetration and target specific inflammatory sites [18,19]. Progress in these domains has facilitated the approval and commercialization of several transdermal drug delivery products, with additional innovations currently in development. In the context of inflammatory diseases, the capacity to deliver drugs directly to inflamed sites—whether joints, skin, or the gastrointestinal tract—offers significant hope for improving therapeutic outcomes.

This article explores the application and progress of TDDSs in the treatment of inflammatory diseases, with a particular focus on their mechanisms of action, advantages, challenges, and future directions. First, it discusses the fundamental principles of transdermal drug delivery, along with the obstacles and pathways involved in drug penetration through the skin. Second, it reviews case studies to examine the developmental progress of TDDSs and highlights the limitations of traditional methods in managing common inflammatory skin diseases. Next, it investigates the advantages of TDDSs over conventional approaches in treating prevalent inflammatory skin diseases, addressing their ability to overcome skin barriers, enhance drug delivery, and improve efficacy. The article also explores the challenges and limitations associated with transdermal drug delivery. Finally, it considers the future prospects of TDDSs, as researchers continue to pursue more effective and less invasive treatment options for inflammatory diseases.

## 2. Barriers to Drug Delivery to Local Skin

Drugs can vary in molecular size, encompassing both micromolecules and macromolecules. Micromolecules typically penetrate the skin with relative ease, whereas macromolecules often require specialized techniques to traverse the skin barrier [20,21]. Some drugs can directly permeate the epidermis, while others must navigate alternative routes, such as hair follicles or sweat glands. Drug penetration occurs through three primary pathways: intercellular, transcellular, and appendageal [22,23,24]. The intercellular pathway is particularly favorable for lipophilic molecules, allowing diffusion between cells; the transcellular pathway involves direct passage through cells; and the appendageal pathway facilitates the absorption of liposomes and nanoparticles via skin appendages. For therapeutic efficacy, medications must be delivered to the affected subcutaneous region at the appropriate concentration and dosage [25].

The skin serves as the principal barrier to drug delivery via local routes. Structurally, the skin comprises two main layers: the epidermis and the dermis. The epidermis, the outermost layer, provides a formidable physical barrier to the penetration of substances. It is primarily composed of keratinocytes, melanocytes, and Langerhans cells. The epidermis can be further subdivided into four distinct layers: the *stratum corneum*, *granular layer*, *stratum spinosum*, and *basal lamina* [26]. The *stratum corneum*, situated at the outermost surface of the epidermis, constitutes the primary barrier function of the skin, permitting only specific molecules—such as lipophilic and low molecular weight drugs—to pass through. This layer is highly cohesive and relatively impermeable [27]. With a thickness ranging from 10 to 15 μm and comprising 15 to 20 layers of keratinocytes, the *stratum corneum* represents a unique biological membrane distinguished by its composition and morphology [28]. It consists of dead keratinocytes embedded within an intercellular lipid matrix. These keratinocytes are bound together by a protein matrix coated with a lipid layer, including ceramides, cholesterol, and other free fatty acids [29]. Together, the *stratum corneum* and granular layer form a highly hydrophobic barrier, impeding the passive penetration of drug molecules exceeding 500 kDa. Consequently, the *stratum corneum*, with its thickness of 15 to 20 μm, acts as a rate-limiting barrier during transdermal drug diffusion [30].

The significant barrier properties of skin tissue are predominantly attributed to the *stratum corneum*, which forms the thin outer layer of the epidermis [31]. This layer is composed of keratinocytes aggregated with lipids organized into multiple layered bilayers. These lipids play a critical role in preventing excessive water loss and blocking the entry of low molecular weight hydrophobic drugs.

Tight junctions, specialized connections between skin cells located in the granular layer, represent the second major barrier to molecular penetration beyond the *stratum corneum*. These junctions regulate the passage of drugs and other substances through the skin by controlling the difficulty of movement via the paracellular pathway (i.e., the space between cells), thereby influencing the efficacy of drug formulations designed to breach the skin barrier. Depending on the size and charge of the molecule, tight junctions can impede the movement of substances from the external environment into the skin’s interior. Certain proteins and peptides can enhance the permeability of this barrier. Ionic permeability is a critical factor affecting both the effectiveness of drug delivery through the skin and the maintenance of skin health. Transepithelial electrical resistance (TEER) can increase ion permeability, and the m19 peptide binds to specific proteins known as claudins (1, 2, 4, and 5) within tight junctions. When m19 attaches to these proteins, TEER is reduced. Similarly, the binding of the 7A5 monoclonal antibody to claudin-1 or the 3B11 monoclonal antibody to claudin-4 can decrease TEER and enhance ion permeability. The AT1002 peptide improves drug delivery efficiency by phosphorylating the tight junction structural protein ZO-1 [25].

Other components of the skin may also limit or decelerate drug diffusion and systemic delivery. The *basement membrane*, located at the interface between the dermis and epidermis, consists of a dense network of structural proteins and carbohydrates and may contribute to the overall barrier system of the skin. Similarly, *blood vessels* beneath this membrane can influence drug transport dynamics.

The *basement membrane* (BM), positioned on the basal side of the basal layer, supports the epidermis and reinforces the skin’s barrier function, hindering the entry of therapeutic drugs. A denser and thicker *basement membrane* may slow the diffusion of larger molecules. Composed primarily of laminins, collagens, proteoglycans, and hyaluronic acid, the *basement membrane* forms a crosslinked, sheet-like structure in conjunction with various other molecules. Research has demonstrated that the thickness of the *basement membrane* in patients with atopic dermatitis is significantly reduced [32].

*Blood vessels* represent the “final barrier” of the skin. Alterations in vascular dynamics can affect the efficacy of transdermal drug delivery. Endothelial cells within *blood vessels* serve as a barrier between the bloodstream and surrounding skin tissue. These cells are highly responsive to various stimuli, particularly increases in pressure or body temperature, which can induce morphological changes and enhance blood flow. When blood flow increases, drugs applied to the skin surface are more effectively absorbed into the circulatory system [33,34,35]. In this section, we discuss the barriers and mechanisms impeding drug penetration across skin layers (Figure 1), using Figdraw2.0 (https://www.figdraw.com/#/), accessed on 25 April 2025.

## 3. Development and Evolution of TDDSs

TDDSs offer an alternative to traditional oral and parenteral administration routes, enabling drug absorption through the skin. The efficacy of these systems is influenced by several factors, including the stratum corneum, pharmacological characteristics of the drug, the carrier medium, temperature, and pH. Given the formidable skin barrier, the fundamental principles governing TDDSs must be meticulously established. In certain instances, the most convenient method of drug administration—the oral route—proves infeasible, necessitating the exploration of alternative pathways. Although intravenous administration can circumvent many limitations associated with oral delivery (such as gastrointestinal and hepatic metabolism), its invasive nature and potential side effects, particularly in the context of chronic administration, underscore the need for additional therapeutic alternatives [36].

TDDSs have evolved through four distinct generations since their inception in 1981 [37]. The first-generation systems are typically employed in conjunction with other penetration-enhancing technologies, though this review does not delve into their specifics.

Second-generation TDDSs focus on disrupting the stratum corneum barrier to facilitate drug passage, commonly utilizing techniques such as iontophoresis, chemical enhancers, and ultrasound [38]. However, these systems are limited to small, lipophilic compounds and remain ineffective for large hydrophilic molecules. Third-generation systems adopt a minimally invasive approach, breaching the stratum corneum to enable the penetration of macromolecular drugs and even vaccines, with methods including electroporation and microneedling [39].

The latest, fourth-generation TDDSs leverage nanocarriers to enhance drug delivery, improving bioavailability and targeting precision while reducing side effects such as inflammation and skin tissue damage at the administration site. Various nanocarriers, including liposomes, ethosomes, and solid lipid nanoparticles, have been developed for this purpose [40].

A significant drawback of early TDDSs is the potential for local irritation and allergic skin reactions during the initial stages of formulation development. Therefore, it is imperative to assess the potential pathogenicity of both drugs and excipients used in these systems [2,41].

### 3.1. Sonophoresis

Sonophoresis, a second-generation transdermal drug delivery technique, employs ultrasound to facilitate the absorption of drugs into the epidermis, dermis, and skin appendages. This method is particularly suited for hydrophilic molecules and macromolecules. Ultrasound generates acoustic waves that induce micro-vibrations within the skin’s epidermis, increasing the kinetic energy of molecules. This technology is most effective at low frequencies and is widely utilized in hospital settings for transdermal drug delivery.

Sonophoresis employs ultrasound to transport drug molecules across the skin. Ultrasonic waves are mechanical waves that propagate longitudinally, parallel to the oscillation of particles. The frequency range of ultrasound used to enhance skin permeability spans from 20 kHz to 16 MHz. Low-frequency ultrasound is particularly effective at penetrating biological tissues with minimal energy attenuation, making it highly suitable for augmenting transdermal drug delivery [42,43,44]. Low-frequency ultrasound has demonstrated significant utility in both industrial and therapeutic applications, such as lithotripsy, liposuction, cataract emulsification, cancer treatment, dental descaling, and ultrasonic surgical tools [45]. These characteristics render low-frequency sonophoresis especially advantageous for drug delivery purposes. Low-frequency sonophoresis is widely applied in cosmetic procedures, skin regeneration, and fat mass repair. The first clinical application of low-frequency sonophoresis involved the delivery of lidocaine liposomes, which significantly reduced the onset time of local anesthesia [46].

Over the years, studies on low-frequency sonophoresis have categorized it into two approaches: simultaneous sonophoresis and preprocessed sonophoresis. Simultaneous sonophoresis enhances transdermal drug delivery through two mechanisms: altering the skin structure to allow drug diffusion and inducing convection via ultrasound. In contrast, preprocessed sonophoresis involves short-term ultrasound application to enhance skin permeability, maintaining elevated permeability over an extended period during which medication is subsequently introduced.

While sonophoresis effectively increases skin permeability, it alone does not provide sufficient efficiency to enhance the penetration of large molecules. Additionally, it may cause irritation, mild stinging, or a burning sensation due to the skin’s sensitivity to temperature variations. Sonophoresis is a time-consuming process, and an intact stratum corneum is essential for effective drug dispersion. Although some drugs can be absorbed at therapeutic doses, sonophoresis alone exhibits low efficiency in facilitating macromolecular penetration.

The precise mechanism of sonophoresis remains incompletely understood. However, it is widely posited that the technique primarily relies on a cavitation mechanism, which enhances drug permeability and penetration through the skin [47]. Other mechanisms, including thermal effects, convective transport, and mechanical effects, have been proposed, though cavitation has garnered significant attention and research focus. As an alternative to oral and subcutaneous administration, sonophoresis—particularly when combined with other drug delivery techniques—is employed for drugs with low bioavailability or those requiring small doses, regardless of molecular weight. Low-frequency sonophoresis-mediated drug delivery demonstrates efficacy in three key areas: vaccine administration, delivery of low-bioavailability drugs, and skin gene delivery. Future challenges for ultrasound-mediated drug delivery include determining the quantity and kinetics of drugs deliverable transdermally, addressing the cost and size of ultrasound equipment, and conducting long-term safety studies, which remain areas requiring detailed investigation. Confirmation of the long-term safety of repeated treatments in the same region is essential. Innovative approaches, such as dual-frequency ultrasound and ultrasound contrast agents, may help resolve permeability limitations [48].

### 3.2. Iontophoresis

Iontophoresis, a second-generation TDDS, utilizes an electric current to deliver substances through skin tissue, overcoming the limitations imposed by the skin barrier. This method typically permits penetration measurements for lipid-soluble substances with molecular weights of approximately 500 Da. A critical requirement for iontophoresis is that drug molecules must ionize into a water-soluble form. This technique enables the delivery of molecules unsuitable for passive transport through the stratum corneum. While the formulation requirements differ markedly from those for lipophilic molecules delivered via traditional transdermal methods, uncharged and water-insoluble molecules can also be adapted for iontophoresis by appropriately modifying their physicochemical properties.

Iontophoresis is particularly effective for enhancing the transdermal delivery of peptides and proteins, achieving this with lower current intensities over short durations. It can be defined as the permeation of ionized drug molecules through biological membranes under the influence of an electric current or as a technique that transports ions or charged molecules into tissues via an electrolyte solution containing the drug, using suitable electrode polarity with a direct or periodic current [49].

Iontophoresis enhances drug delivery across the skin through two primary mechanisms: electrical repulsion and electroosmosis. Electrical repulsion involves the application of an electric field to a charged permeable entity, driving it through the skin. The second mechanism, electroosmosis, arises from the skin’s inherent net negative charge at physiological pH levels [50,51]. This technology has been developed for numerous drugs with poor transdermal permeability, including high-molecular-weight electrolytes and challenging-to-administer proteins, peptides, and oligonucleotides. It also holds considerable potential for delivering charged peptides as therapeutic agents. However, while iontophoresis significantly improves the transdermal absorption of many drugs, it has yet to demonstrate substantial penetration of larger peptides, such as insulin [52].

Although numerous potent enhancers have been identified, their clinical utility is constrained by toxic side effects. Unlike chemical and physical penetration enhancement techniques that may impair skin barrier function, recent advancements suggest that iontophoresis, as a physical enhancement method, offers a preferable alternative. Chemical enhancers, such as surfactants and oils, denature keratinocytes or disrupt lipid bilayers, while physical enhancers—including iontophoresis, electroporation, thermal ablation, liquid jet injectors, and sonophoresis—provide diverse mechanisms to improve drug penetration.

### 3.3. Chemical Penetration Enhancer

The use of chemical penetration enhancers represents one of the most widely adopted strategies within second-generation TDDSs. Owing to its distinct advantages—such as ease of application and high compatibility—it is a preferred method for improving drug penetration through the skin. Chemical penetration enhancers (CPEs) are extensively studied in TDDSs, offering the following benefits: (1) simple application [53], (2) non-invasive administration [54], (3) excellent compatibility with diverse formulations [55,56], and (4) relatively low cost [57]. S. Mojtaba Taghizadeh et al. investigated the effects of various chemical penetration enhancers on the skin penetration flux, viscosity, and peel strength of buprenorphine transdermal patches, finding that both skin penetration flux and adhesive performance were modulated by the concentration of each enhancer [58]. Shah et al. propose that enhancers operate through the following mechanisms: (1) increasing the diffusion rate of drugs within the skin, (2) fluidizing the stratum corneum to reduce its barrier function, (3) elevating the thermodynamic activity of drugs within the carrier, and (4) altering the drug’s partition coefficient.

CPEs enhance drug penetration by disrupting the stratum corneum barrier, though this can lead to adverse reactions, such as skin irritation and allergies. These reactions may ultimately trigger inflammation via various cytokines, making the mechanisms by which CPEs compromise the stratum corneum barrier critical for balancing efficacy and safety. To enhance the safety of CPEs, it is advisable to avoid lipid extraction from the stratum corneum or modification of stratum corneum protein mechanisms. For some CPEs, safety concerns escalate with increasing concentrations [59].

Current research progress indicates that optimizing the compatibility between CPEs and drugs can enhance skin penetration efficiency. The combined use of CPEs improves efficacy and mitigates skin risks by facilitating drug penetration, though selecting appropriate CPE combinations remains challenging. Compared to traditional CPEs, newer variants exhibit higher skin penetration efficiency, with some capable of enhancing macromolecular permeability. While most CPEs have improved skin permeability efficiency, the risks and applicability of novel CPEs across different drug types warrant further investigation [59]. Despite the use of approximately 150 chemical penetration enhancers, their effectiveness and safety profiles have limited their application in TDDSs. In the pharmaceutical industry, issues related to toxicity, allergies, hypersensitivity, and sensitization highlight the need for further refinement.

### 3.4. Electroporation

The stratum corneum barrier poses a significant challenge to the advancement of TDDSs. Electroporation, a third-generation technique, employs short electrical pulses to create transient pores in the skin, facilitating drug entry. This method is particularly favored for delivering large molecules, including genetic material. Compared to iontophoresis, electroporation utilizes high-voltage pulses (50–500 V) applied over brief durations to disrupt the lipid bilayer structure of the stratum corneum, thereby forming temporary pathways that enhance transdermal drug absorption and alter lipid tissue organization [60]. The efficiency of transdermal permeability induced by electroporation depends on formulation parameters and the physicochemical properties of the drug molecules [61]. Drug transport increases with parameters such as pulse voltage, duration, and rate of electrical stimulation. Xiao Chen et al. demonstrated that electroporation generates reversible channels or pores in the stratum corneum, with penetration effects persisting for over 12 h, providing a basis for optimizing electroporation parameters for various drugs’ transdermal absorption and offering a reference for efficient penetration comparisons [60]. Electroporation offers several advantages: (1) adjustable parameters to control transdermal penetration rate and extent, (2) reversible pores with minimal damage post-high-voltage pulses, and (3) enhanced penetration for most drugs, including high-molecular-weight hydrophilic drugs and biomolecules such as insulin and DNA [62]. The enhanced transdermal absorption mechanism of electroporation is generally attributed to high-voltage pulsed electric fields disrupting the ordered structure of the stratum corneum, resulting in reversible permeation pores [63]. Electrochemotherapy, a successful clinical application of electroporation, is used to treat skin tumors in patients with malignant melanoma.

Electroporation holds promising prospects for expanding the scope of transdermal drug delivery to biotechnology products and other hydrophilic macromolecules. However, its progress is hindered by insufficient transmission efficiency compared to iontophoresis and a lack of comprehensive chronic toxicity data. Nevertheless, electroporation remains an attractive and competitive drug delivery technology [64].

### 3.5. Microneedling

Microneedling and microneedle patches represent distinct third-generation transdermal drug delivery systems (TDDSs) [65]. Microneedling is a minimally invasive dermatological procedure that employs devices such as dermarollers or motorized pens equipped with fine needles to create controlled micro-injuries in the skin [66]. The primary objective of microneedling is to stimulate collagen production and improve skin texture, making it a popular treatment for scars, wrinkles, and other dermatological conditions [67]. Notably, in microneedling, the drug is not delivered through the needles themselves but is applied topically post-procedure to penetrate the skin via the created microchannels [68]. In contrast, microneedle patches are advanced transdermal delivery systems composed of arrays of micron-scale needles that directly deliver therapeutic agents into the skin [69]. These patches are designed for self-administration and can be categorized into several types based on their structure and function: solid microneedles, coated microneedles, dissolving microneedles, hollow microneedles, and hydrogel-forming microneedles [70]. Unlike microneedling, microneedle patches deliver drugs through the needles themselves, enabling precise dosing and targeted delivery to specific skin layers [71]. This method offers advantages such as improved patient compliance, reduced pain, and the potential for self-administration [72]. Microneedle patches have been explored for the delivery of various therapeutics, including vaccines, insulin, and other macromolecules [73]. In summary, while both techniques involve micro-scale skin penetration, microneedling primarily serves as a mechanical method to enhance the absorption of topically applied agents, whereas microneedle patches function as direct drug delivery systems. Recognizing and articulating this distinction is crucial for accurately conveying the mechanisms and applications of these technologies in transdermal drug delivery research.

#### 3.5.1. Solid Microneedles

Solid microneedles operate by increasing skin permeability through the creation of microchannels, followed by the application of drug patches over these channels. Drugs are delivered via passive diffusion, and the microchannels must close after needle removal to prevent the ingress of toxic substances or pathogenic microorganisms [74]. The rectangular cup-shaped silicon microneedle array developed by Vinayakumar et al. demonstrates potential to reduce drug leakage, enhance delivery efficiency, and enable simultaneous transdermal delivery of multiple drugs [75]. A limitation of solid microneedles in TDDSs is the complexity of the delivery route, typically requiring a two-step process to administer the drug to the skin.

Solid microneedles can be fabricated from polymers. Olatunji et al. prepared microneedles from polymer films extracted from tilapia scales, successfully inserting them into pig skin using methylene blue as a model drug [76]. Monika Kaur et al. utilized microneedle rollers and stainless steel microneedles for the transdermal delivery of verapamil hydrochloride and amlodipine besylate, finding that both significantly increased delivery rates [77]. Stainless steel microneedle arrays (750 μm) manufactured by Microneedle Systems^®^ markedly improved the transdermal delivery rates of captopril and metoprolol tartrate [78]. Solid microneedles are also effective for vaccine administration, eliciting more persistent and robust antibody responses compared to intramuscular injections, thereby enhancing vaccine efficacy.

#### 3.5.2. Dissolved Microneedles

Dissolved microneedles are highly favored due to their “one-step” operation, offering convenience to patients. Their manufacturing principle, termed “puncture and release”, involves microneedles made from polysaccharides or other polymers that dissolve upon use, releasing encapsulated drugs into the skin. Micromoulding is the preferred method for fabricating dissolved microneedles. As some drugs and vaccines are heat-sensitive, they are typically incorporated into a solution with drug extracts, filled into a mold, and dried under mild conditions. The manufacturing process entails pouring a polymer solution into a negative mold, filling the mold’s microcavities under vacuum or pressure, and drying via ambient conditions, centrifugation, or pressure [79,80,81].

Jiaxin Li et al. designed PVA microneedles loaded with F-BPA as a transdermal delivery system for boron neutron capture therapy (BNCT). The dissolution of these microneedles enhanced BNCT delivery efficiency and demonstrated potent therapeutic effects against melanoma [82]. Yu Shuai Wang et al. reported that a 1:1 mixture of arbutin (Arb) and vitamin C (Vc) exhibited the greatest inhibition of melanin production and tyrosinase activity in B16 mouse melanoma cells. This hyaluronic acid (HA)-based dissolved microneedle array (DMNA) overcomes skin barriers, improves local drug delivery, and prevents potential skin issues associated with pigmentary diseases [83].

#### 3.5.3. Hydrogel Microneedles

Hydrogel microneedles are easily sterilized and can be completely removed from the skin. They can be fabricated from super-swelling or crosslinked polymers. The hydrophilic nature of these polymers allows them to absorb substantial amounts of water into their three-dimensional structure. Upon insertion into the skin, the presence of tissue fluid causes these polymers to swell, forming channels between capillary circulation and drug patches. Initially, their role is to breach the skin barrier; subsequently, during dissolution, they regulate the rate of drug release across the cell membrane, offering flexibility in size and shape [84].

Ismaiel A. Tekko et al. reported a novel hydrogel-forming microneedle array (HFMN). The integrated patch produced by this array represents a promising minimally invasive TDDS capable of overcoming the skin barrier and delivering methotrexate and other drugs continuously. Although methotrexate is cytotoxic, this hydrogel array can be fully removed from the skin, resulting in only mild erythema [85].

#### 3.5.4. Coated Microneedles

Coated microneedles refer to microneedles overlaid with drug-containing dispersions. The tips of these microneedles are coated with drugs using methods such as immersion, gas jet drying, inkjet printing, or spray coating [86]. A method employing Electrohydrodynamic Atomization (EHDA) principles was reported for preparing smart microneedle coatings, yielding nano- and micro-scale drug coatings [87]. The drug delivery mechanism of coated microneedles, termed “coating and puncture”, involves inserting the microneedle patch into the skin and releasing the drug applied to the needle tips [88]. Coated microneedle patches exhibit significant potential for vaccine delivery, eliciting prolonged and robust antibody responses [89].

#### 3.5.5. Hollow Microneedles

Hollow microneedles (HMs) resemble conventional subcutaneous needles but are shorter, with drug liquid formulations injected through apertures in the microneedles. HMs can be manufactured using commercially available 30-gauge subcutaneous injection needles [90]. Yuzhakov et al. utilized a UV excimer laser to fabricate elongated holes (diameter: 10 μm, depth: 1 mm) with an aspect ratio of 100 in biodegradable polymer materials. These holes, distributed along the microneedle centerline, successfully drew plasma into the cavities via capillary forces [91]. Lyon et al. reported a method for fabricating hollow microneedles using a composite of vertically aligned carbon nanotubes and polyimide [92].

Maintaining a sufficient and stable flow rate during transdermal drug delivery, without compromising the mechanical strength of HMs, is critical. The compression of dense skin tissue at the needle tip during penetration is a primary factor affecting delivery efficiency [93]. Pressure and flow rate can be adjusted in HMs to enable rapid injections, slow infusion, or variable delivery rates [70]. HMs are suitable for administering larger doses of drug solutions [93]. Offsetting sharp microneedles from the center or side of the hole can enhance flow velocity. Various strategies, such as increasing the number of microneedle holes or coating them with metal, can elevate flow rates, though these may reduce needle sharpness [94].

HMs bypass the stratum corneum, delivering compounds directly to the epidermis or dermis. This approach is particularly effective for high molecular weight complexes, such as proteins, oligonucleotides, and vaccines [27]. By precisely targeting mRNA sequences and designing complementary antisense agents, specific and effective gene silencing for certain diseases can be achieved. Downstream regulation via antisense technology employs drugs such as antisense oligonucleotides, ribozymes, short interfering RNA (siRNA), microRNA (miRNA), and aptamers to manipulate gene expression and treat various conditions [95].

The ideal hollow microneedle exhibits sufficient mechanical strength and prevents hole blockage during transdermal delivery. Further research is needed to refine hollow microneedle technology to establish it as the “gold standard” for drug delivery systems.

### 3.6. Transdermal Drug Delivery Using Nanocarriers

In recent years, TDDSs employing nanocarriers have emerged as a fourth-generation approach to minimize skin tissue damage. Nanocarriers are minute carriers designed to facilitate the delivery of drugs that struggle to penetrate the skin. Hydrophilic biomolecules, which are typically water-attracted and difficult to transport through the skin, benefit significantly from nanocarriers, which excel in delivering such substances [40].

#### 3.6.1. Liposomes

Liposomes, among the most established nanocarriers, are one of the few nanoformulations utilized in clinical settings, with particle sizes ranging from 50 to 1000 nanometers [96]. Their unique lipid bilayer structure enables the encapsulation of water-soluble drugs within the core and lipid-soluble drugs between the bilayers [97,98,99]. The amphiphilic, biocompatible, and biodegradable properties of liposomes are highly valuable for delivering traditional Chinese medicines. Moreover, these carriers enhance therapeutic efficacy and safety by improving bioavailability, enabling sustained release, and facilitating localized drug delivery [100].

#### 3.6.2. Lipid Nanoparticles

Lipid nanoparticles are categorized into solid lipid nanoparticles (SLNs) and polymer nanoparticles (PNPs). SLNs, prepared from solid natural or synthetic lipids, typically range from 50 to 1000 nanometers in size. In recent years, SLNs have gained prominence due to their high drug-loading capacity, broad applicability, controlled release properties, favorable biosafety, and stability [101,102]. PNPs, with particle sizes spanning 10 to 1000 nanometers, are formulated from biocompatible and biodegradable polymers [103,104,105]. They represent a promising drug delivery tool, offering a straightforward manufacturing process, targeted delivery, enhanced safety, increased therapeutic efficacy, and reduced side effects [106,107,108]. Liquid crystal nanoparticles (LCNs) loaded with tacrolimus exemplify direct skin delivery, enhancing penetration through the skin barrier and improving psoriasis treatment outcomes [109].

#### 3.6.3. Bicelles

Hema Chaudhary et al. developed a stable, reproducible, and patient-friendly TDDS termed “Nanocarrier Transdermal Gel” (NCTG), which is non-irritating to the skin and effective in treating skin inflammation. This system’s efficacy may stem from the encapsulation of diclofenac diethylamine (DDEA) and curcumin (CRM) in elastic nanocarrier vesicles, reducing the likelihood of contact dermatitis [110].

#### 3.6.4. Nanoemulsions

Nanoemulsions are stable liquid systems with droplet sizes ranging from 10 to 100 nm. Compared to traditional formulations, they enhance solubility, absorption, and diffusion, promote localized administration, bypass first-pass metabolism and degradation, and are particularly effective for delivering cyclosporine A, estradiol, and lidocaine [111]. Nanoemulsions have proven effective against inflammation, anesthesia, antifungal, and steroid drugs. Chuanqi Wang et al. revealed that PTX-COUP outperformed PTX in suppressing tumor growth, offering new perspectives and strategies for the development of innovative targeted drug delivery systems [112]. Chaozheng Zhang et al. developed a dual-targeting nanomedicine, FA-HA-SS-PPT, through a self-assembly strategy. The resulting FA-HA-SS-PPT nanoparticles leveraged the specific binding affinities toward both folate receptors (FRs) and CD44 receptors to achieve targeted drug delivery into tumor cells. Moreover, the nanoparticles facilitated a self-cycling release of PPT triggered by the high intracellular glutathione (GSH) concentrations. This work offers a promising, safe, and effective strategy for the treatment of tumors characterized by the overexpression of multiple receptors through the use of multi-targeted nanomedicines [113]. Unlike macroemulsions, nanoemulsions excel in droplet formation and stability, capable of dissolving both hydrophilic and hydrophobic drugs while reducing diffusion barriers to drug penetration. Nanoemulsion-based hydrogels integrate viscosity and biocompatibility, with gelling agents such as xanthan gum and carbomer enhancing performance. These hydrogels offer prolonged drug delivery and high loading capacity. Microemulsions further improve drug solubility and reduce toxicity [114,115]. In this section, we discuss the schematic representation of nanoemulsion-based transdermal drug delivery system: formulation components, skin penetration mechanisms, and therapeutic outcomes (Figure 2), using Figdraw2.0 (https://www.figdraw.com/#/) accessed on 7 May 2025.

## 4. Application of Transdermal Drug Delivery Systems in Inflammatory Skin Diseases

### 4.1. Overview of Inflammatory Skin Diseases

Inflammatory skin disease is a form of skin inflammation that may arise from infections caused by bacteria, viruses, or fungi, affecting approximately 20% to 25% of the population [116]. The skin barrier, innate immunity, and adaptive immunity are critical components of skin defense, each potentially contributing to the development of inflammatory skin diseases. They detect antigens entering through the skin, activate and differentiate T cells, produce cytokines, and interact with other immune cells, ultimately signaling the central immune system to respond to potential threats [117]. Cytokines play a decisive role in shaping T cell responses, influencing their differentiation, activation, and overall immune activity in the context of inflammatory skin diseases. Their contributions to promoting inflammation and altering keratinocyte behavior underscore their significance in the pathogenesis of these conditions [118].

Early research suggested a higher incidence of inflammatory skin diseases among ethnic minority groups. However, some studies have indicated that, compared to other populations, ethnic minority groups may have a lower likelihood of developing these conditions [116]. Recent research highlights that inflammatory skin diseases are prevalent among the elderly, affecting one-eighth of U.S. residents aged 50 and older [119]. A meta-analysis examining the epidemiological characteristics of seborrheic dermatitis reported that adults exhibit a higher incidence rate than children and newborns, with the prevalence being highest in South Africa and lowest in India [120]. Atopic dermatitis typically emerges within the first few years of life, often resolving in over 70% of cases by school age. In some developed countries, the incidence of atopic dermatitis in children has risen to 20%. Compared to urban areas, rural populations exhibit a lower incidence rate, and Eastern European countries report lower rates than Western European countries [121]. Psoriasis patients face a high incidence of comorbidities, including diabetes, coronary heart disease, and depression [122].

Common medications for relieving skin inflammation include corticosteroids that reduce swelling and redness, vitamin D analogues that slow skin cell growth, and calmodulin inhibitors that lower the skin’s immune response [123]. Local therapies include emollients (moisturizers), topical steroid creams, antihistamines, and biologic therapies. Moisturizing agents help maintain skin hydration and repair the skin barrier [124,125]. However, overuse may lead to skin thinning [126], with a skin penetration rate of only 10–20% [2]. Prolonged use can also cause serious side effects, including metabolic disorders and immune suppression [7,127]. Atopic dermatitis, psoriasis, and acne are among the most common inflammatory skin diseases.

AD is a complex disease with symptoms varying among patients. Its appearance and sensation differ from person to person, rendering treatment of AD far from a one-size-fits-all scenario. An approach combining good efficacy with minimal side effects has yet to be established. Considering these factors, developing a treatment plan that effectively controls the condition for all patients remains a priority.

Acne, another prevalent inflammatory skin disease, arises from increased sebum secretion, irregular peeling, proliferation of acne-causing bacteria, and inflammation [128]. Treatment methods for acne target these four pathogenic factors and include local, systemic, natural product-based, and non-pharmacological approaches. Combination therapies addressing multiple acne mechanisms are typically more effective [129]. Topical corticosteroids are more effective for non-inflammatory acne. Oral progesterone drugs are effective for treating acne related to hormonal imbalances in females. Diversified treatments, such as oral prednisone, can address inflammatory acne [130]. However, prolonged antibiotic use leads to resistance in acne bacteria, diminishing their effectiveness over time. The emergence of natural medicines may help mitigate antibiotic resistance, though their efficacy requires further clinical validation [131]. Currently, effective natural drugs for acne include green tea, minerals, resveratrol, seaweed extract, and tea tree oil [132]. These natural medicines can inhibit acne bacteria growth and exhibit antibacterial or anti-inflammatory effects [133]. Physical therapies, such as comedone extraction, cryotherapy, electrocautery, and phototherapy, are also employed. Combination therapy can enhance patient compliance and treatment effectiveness, reducing antibiotic dosages and improving adherence when paired with systemic treatments. Given widespread antibiotic resistance, many patients exhibit stronger adherence to natural therapies, seeking more comfortable and efficient treatment options.

Psoriasis, a chronic inflammatory skin disease, can be managed with local treatments for mild to moderate cases. Corticosteroids and other anti-inflammatory ointments help reduce swelling and redness. In psoriasis, skin cell growth is typically excessively rapid, and vitamin D analogues slow this process, alongside phototherapy [134]. Moderate to severe psoriasis often requires systemic treatment with oral or injectable medications, primarily small-molecule drugs such as methotrexate, cyclosporine, apremilast, fumaric acid esters (FAE), and acitretin. Psoriasis involves rapid skin cell proliferation, and slowing DNA synthesis can help control the condition. Methotrexate blocks the production of thymine and purine to prevent DNA synthesis in the body [135]. However, its therapeutic effect is limited by low solubility and poor absorption. As an autoimmune disease, psoriasis benefits from cyclosporine, a drug that suppresses the immune system. Cyclosporine works by blocking specific immune pathways that drive inflammation and skin cell growth. It can rapidly control psoriasis symptoms and may be used for maintenance therapy for up to two years, helping prevent recurrence after symptom improvement [136]. Apremilast, a phosphodiesterase 4 inhibitor, prevents the breakdown of cyclic AMP (cAMP) by phosphodiesterase 4. Elevated cAMP levels help alleviate inflammation by reducing pro-inflammatory cytokines such as TNF-α, IFN-γ, and IL-12 while increasing the anti-inflammatory cytokine IL-10. Apremilast exhibits broad anti-inflammatory effects on keratinocytes, fibroblasts, and endothelial cells [137,138]. FAE, small compounds with unique properties, enhance the immune system and reduce inflammation [139]. Acitretin, a retinoid related to vitamin A, affects skin cell growth and development by interacting with specific nuclear receptor proteins, normalizing keratinocyte growth and maturation [140,141]. However, small-molecule drugs often carry significant side effects. Methotrexate may cause nausea, leukopenia, elevated liver enzymes, and teratogenicity. Cyclosporine can lead to hypertension, kidney damage, and an increased risk of certain non-melanoma skin cancers. FAE may induce stomach discomfort, skin flushing, and leukopenia. Apremilast can elevate other inflammatory signals, resulting in gastrointestinal issues and upper respiratory tract infections. Common side effects of acitretin include cheilitis, conjunctivitis, hepatitis, and teratogenicity. Biopharmaceuticals, such as monoclonal antibodies and receptor fusion proteins, represent emerging therapies targeting specific immune pathways, including the IL-23/Th17 axis and tumor necrosis factor-alpha signaling, to reduce inflammation and skin cell growth [142]. Inflammation in psoriasis causes the body to overreact to harmful substances, subsequently leading to skin complications.

### 4.2. Application of Transdermal Drug Delivery System in Inflammatory Skin Diseases

The skin barrier, the protective outer layer of our skin, prevents moisture loss and the entry of harmful substances. Skin barrier dysfunction and inflammation interact in a cyclical manner [143]. A weakened skin barrier can trigger inflammation, which further damages the barrier. This cycle can perpetuate and exacerbate the condition over time [144]. Consequently, when a TDDS strengthens the skin barrier, it reduces the severity of inflammation as the barrier recovers. Simultaneously, TDDS enhances the skin penetration rate of drugs, bypasses first-pass metabolism, promotes consistent absorption, increases drug utilization, reduces side effects, achieves superior therapeutic outcomes, and improves patient compliance, thereby more effectively managing inflammation. Additionally, TDDS can minimize the amount of drug entering the superficial capillaries of the dermis after traversing skin tissue, further reducing side effects. Various studies have demonstrated that TDDS holds broad prospects and excellent patient compliance in treating inflammatory skin diseases such as psoriasis, atopic dermatitis, and acne. As of December 2024, we conducted a comprehensive literature search on *PubMed* using the advanced search function with the following query: (Transdermal Drug Delivery System[Title/Abstract]) AND (((Psoriasis[Title/Abstract]) OR (acne[Title/Abstract])) OR (atopic dermatitis[Title/Abstract])). We systematically reviewed all relevant publications on the application of TDDSs in inflammatory skin diseases. An updated search using the same strategy was performed in May 2025 to verify and supplement the initial results. In total, 22 relevant articles were identified. This section presents the applications of TDDSs in the treatment of inflammatory skin diseases.

Rana Obaidat et al. developed a specialized nanofiber (NF) microfiber system to deliver pioglitazone (PGZ) through the skin [145]. Their research revealed that, compared to cast thin films, nanofibers increase pioglitazone flux fivefold due to their larger surface area, which provides greater space for drug absorption. Moreover, pioglitazone tends to remain in the skin layers, reducing systemic drug levels and minimizing side effects while treating skin conditions.

Ji Heun Jeong et al. developed a pullulan hydrogel incorporating *Rhus verniciflua* extract (RVE), which exhibits antipruritic and anti-inflammatory effects [146]. This study highlighted the potential of hydrogels as a drug delivery system for atopic dermatitis. The *Rhus verniciflua* extract-loaded pullulan hydrogel demonstrated dual functions: water supplementation and sustained drug delivery [146]. Wenyi Wang et al. formulated a P407/CMCS composite thermosensitive hydrogel as a gel-based TDDS for atopic dermatitis treatment [147]. Carboxymethyl chitosan (CMCS) enhances the porous structure of the P407/CMCS hydrogel, facilitating drug release. CMCS adjusts the sol-gel transition temperature without altering the hydrogel’s properties. This hydrogel maintains skin hydration to alleviate the dry skin symptoms of atopic dermatitis patients while delivering drugs directly to the skin. The P407/CMCS hydrogel exhibits favorable transdermal characteristics and safety, offering a promising dual-function therapy for atopic dermatitis. Yuan Yang et al. developed a conductive transdermal drug delivery system (c-TDS) that enables on-demand drug loading and release via electrical stimulation. By controlling the charge and thickness of the polypyrrole (pPy) film, the system improves the drug release quantity and rate. It effectively treats atopic dermatitis-related skin inflammation by reducing inflammatory cell infiltration and factors [148]. With a strong drug-loading capacity, this system can deliver various drug types and may integrate with biosensors in smart drug delivery systems, advancing the development of intelligent microneedle patches for disease management. Jeong Hae Choi et al. found that low-temperature atmospheric pressure plasma (LTAPP) enhances the delivery of Jaun ointment (JO), transporting anti-inflammatory components, improving atopic dermatitis symptoms, and reducing dermatitis scores [149]. The primary constituents of JO, Angelica gigas and Lithospermum erythrorhizon, are recognized for their anti-inflammatory properties, though their efficacy is limited by poor skin penetration. LTAPP enhances transdermal delivery, amplifying JO’s anti-inflammatory effects, alleviating atopic dermatitis symptoms, and modulating immune responses. Research indicates that LTAPP-JO treatment inhibits key proteins involved in atopic dermatitis, enhances JO absorption without tissue damage, reduces eosinophil counts, and lowers pro-inflammatory cytokine levels in the skin, improving localized treatment outcomes. The study also underscored the role of NF-κB in atopic dermatitis and the potential of LTAPP-JO as a therapeutic option. Triptolide, known for its anti-inflammatory, analgesic, antitumor, and immunomodulatory properties, was formulated by Meng Yang et al. into a triptolide nano-lotion gel. This system replenishes keratin, alters the stratum corneum structure by disrupting it, modifies lipid arrangements, and moisturizes keratin, collectively enhancing skin drug absorption. It also mitigates adverse digestive, urinary, and reproductive system reactions associated with oral triptolide administration [150]. The triptolide nano-lotion gel holds promise for treating dermatitis with reduced toxicity.

Psoriasis, a chronic inflammatory skin disease, thickens the skin and produces scales, hardening the stratum corneum and impeding drug penetration and efficacy. Transdermal patches facilitate drug penetration into the skin. Polymers are essential for the mechanical properties and controlled release of transdermal patches. Carboxymethyl cellulose sodium (CMC-Na) and hydroxypropyl methylcellulose (HPMC) are key polymers with swelling and viscosity properties. Incorporating hydrophilic polymers like CMC-Na, HPMC, or ethyl cellulose into transdermal patches significantly influences drug release and penetration. These polymers form a gel-like layer upon skin contact, promoting better drug absorption. High concentrations of CMC-Na have been shown to enhance drug release and skin penetration. Muhammad Shahid Latif et al. demonstrated that transdermal patches made of CMC-Na and HPMC effectively deliver methotrexate to the skin, alleviating psoriasis-induced skin inflammation [151]. They also developed a methotrexate-containing patch using HPMC and ethyl cellulose for psoriasis treatment [152]. Huaiji Wang et al. highlighted a hyaluronic acid (HA)-based microneedle patch (HM/MN patch) loaded with methotrexate nanoparticles. This patch reduces skin thickening in psoriasis, lowers IL-6 and TNF-α levels in the body, and mitigates inflammation more effectively than traditional methods, targeting the immune system with fewer side effects [153]. Yongjian Song et al. developed a thermoresponsive hydrogel derived from ionic liquid microemulsion (IL microemulsion) for transdermal methotrexate delivery in psoriasis treatment [154]. This innovative formulation retains the thermoresponsiveness of IL microemulsion while enhancing adhesion and mechanical properties. The IL microemulsion increases methotrexate solubility ninefold and permeability sixfold, exhibiting good antibacterial activity, no hemolysis, and excellent biocompatibility. The microemulsion gel enhances skin affinity, strengthens formulation adhesion, improves bioavailability, targets drug delivery, and extends release characteristics. It effectively treats imiquimod-induced skin inflammation without side effects. Eman Zmaily Dahmash et al. achieved high drug capture and loading efficiency with thymoquinone polyamide-based arginine (TQ-Arg-PA) nanocapsules, offering a potential alternative to traditional therapies like corticosteroids [155]. Aloe vera in transdermal patches enhances thymoquinone flux, suggesting its utility in psoriasis treatment. The nano-transdermal delivery system combining 9-ol and TCeO_2_ nanoparticles effectively treats psoriasis [156]. It integrates TRA to prevent excessive keratinocyte growth and TCeO_2_ to target mitochondria, reducing reactive oxygen species (ROS) levels and inflammation. This combination effectively reduces inflammation and oxidative stress. Zi-Ying Zhan et al. showed that parthenolide alleviates psoriatic symptoms and regulates keratinocyte behavior, with transdermal administration outperforming oral delivery [157]. Parthenolide mitigates in vitro skin inflammation by inhibiting NETs, targeting the IL-36 signaling pathway—a key intervention point in psoriasis treatment—thereby improving psoriasis-like skin inflammation. These findings support further exploration of parthenolide as a potential psoriasis therapy. Yulin Hua et al. synthesized a reactive oxygen species-sensitive material, methoxypolyethylene glycol thioether thiol (mPEG-SS), using mPEG as the base structure with sulfur modifications. The mPEG-SS-calcipotriol (mPEG-SS-CPT, PSC) nanomicelle TDDS encapsulates calcipotriol, offering a promising approach to treat psoriasis by targeting oxidative stress from ROS [158]. Calcipotriol (CPT), a key topical drug, regulates keratinocyte behavior. ROS accumulation in psoriasis promotes inflammation and keratinocyte proliferation, inducing inflammatory factors, DNA modification, and oxidative damage. The mPEG-SS material enhances biocompatibility and solubility in drug delivery systems. Laurent L’homme et al. discovered that an atromentin-1-derived peptide protein tyrosine phosphatase (AP-PTPP) conjugate significantly reduces the severity of inflammatory skin diseases, alleviating skin thickening and inflammation in conditions like contact dermatitis and psoriatic dermatitis [159]. Hyun Jeong Ju et al. demonstrated that dissolving microneedle patches can enhance transdermal drug delivery and improve therapeutic outcomes in psoriatic plaques, particularly those resistant to conventional topical treatments, thereby alleviating cutaneous inflammation [160]. Chaoxiong Wu et al. developed a transdermal drug delivery system capable of effectively modulating reactive oxygen species (ROS) levels within the psoriatic microenvironment, offering a promising in situ strategy for mitigating inflammation associated with psoriasis [161].

Acne, a multifactorial disease linked to sebum production, keratinocyte proliferation, and androgens, affects 85% of individuals aged 11–30. Oxidative stress from reactive oxygen species (ROS) can contribute to acne [162,163]. Sopan Nangare et al. developed a TDDS using a transfer gel formulation of mulberry leaf extract for acne treatment, demonstrating excellent skin permeability and antioxidant activity [164]. Qi Wang et al. developed a high-molecular-weight hyaluronic acid (HA)-based microneedle system enriched with eugenol, combining eugenol delivery with photothermal therapy (PTT) for effective acne treatment. Eugenol-loaded HA-based dissolving microneedles (E@P-EO-HA MNs) efficiently deliver eugenol and exhibit strong photothermal properties. They demonstrate robust antibacterial activity against acne-causing bacteria, promote sebaceous gland atrophy, reduce inflammation, and aid skin healing [165]. In this section, we discuss the Recent advances in TDDS for the treatment of ISD (Table 1), using Word.

### 4.3. Limitations of Transdermal Drug Delivery System Under Inflammatory Conditions

High doses or frequent use of TDDSs can lead to skin irritation. Although TDDSs can mitigate some systemic side effects, they may still induce skin-related issues, such as atrophy, cytotoxicity, and phototoxicity [166,167]. Prolonged use may also result in dermatitis triggered by allergic reactions. Allergic contact dermatitis associated with TDDSs is uncommon, with approximately 8% of patients requiring discontinuation due to skin reactions. This suggests that while allergic responses are infrequent, their severity typically ranges from mild to moderate [168]. Remedios Pérez-Calderón et al. reported a case of a 76-year-old man who developed a generalized eczematous rash after using a nitroglycerin transdermal system for one year [169]. The authors emphasize the need to raise awareness of potential allergens in transdermal patches, noting that any component—active drugs, adhesives, or excipients—may provoke allergic reactions. Although transdermal systems are generally well tolerated, individual sensitivities can lead to severe allergic responses. Consequently, healthcare professionals must recognize the potential allergens within these systems, and a thorough understanding of a patient’s medical history is essential for managing skin reactions to medications.

The stratum corneum, pharmacological properties, carrier medium, and skin temperature and pH of the host organism all influence the transdermal penetration efficiency of TDDSs, potentially reducing their therapeutic efficacy [37]. Researchers should advocate for the advancement of TDDSs and focus on refining transport carriers to improve penetration efficiency.

TDDSs also encounter numerous challenges in regulation and manufacturing. Transitioning from laboratory-scale production to large-scale manufacturing presents significant difficulties. The high costs associated with developing TDDS products—including expenses for research, production, and regulatory compliance—may impede the progress of new TDDS innovations. The development and commercialization of TDDS products are subject to stringent oversight by governmental authorities, which can delay the introduction of new products to the market [170]. Extensive clinical trials are required to validate the effectiveness and safety of TDDS products, and ensuring reproducibility and consistency across production batches is critical to prevent variations in pharmacological effects. TDDSs are sensitive to environmental factors, such as light, temperature, and humidity. Maintaining stable quality, temperature, humidity, and sterile conditions during production, packaging, and storage is imperative to prevent the degradation of drug efficacy [171]. Currently, there is no standardized method for assessing the bioequivalence of TDDS products. These challenges underscore the complexity of TDDS development and highlight the need for ongoing research and innovation [172].

## 5. Limitations and Future Prospects

The TDDS offers substantial advantages in treating inflammatory skin diseases by directly delivering drugs to the site of skin lesions. This targeted approach enhances therapeutic efficacy while minimizing systemic exposure, which is particularly advantageous for conditions such as psoriasis, atopic dermatitis, and acne. TDDSs prioritize improving drug penetration through the skin barrier, with technologies like microneedles and electroporation facilitating deeper drug delivery. Various carrier systems have enhanced drug bioavailability. The latest generation of TDDSs employs nanocarriers to more effectively transport hydrophilic biomolecules to the delivery site, thereby improving the efficiency of anti-inflammatory agent administration. The painless application method increases patient compliance, proving beneficial for the long-term management of chronic inflammatory skin diseases. The potential of TDDS in treating inflammatory skin diseases is promising; however, most related research remains in the preclinical stage. Consequently, further clinical trials are essential to enhance the safety and effectiveness of TDDSs.

## 6. Conclusions

The application of TDDSs in managing inflammatory diseases marks a significant advancement in pharmaceutical technology, offering patients more effective, convenient, and safe treatment options. Despite existing challenges, the progress achieved in this field suggests that TDDSs have the potential to fundamentally transform the management of inflammatory diseases, providing patients with improved efficacy, convenience, and safety in their treatment regimens.

## Figures and Tables

**Figure 1 pharmaceutics-17-00746-f001:**
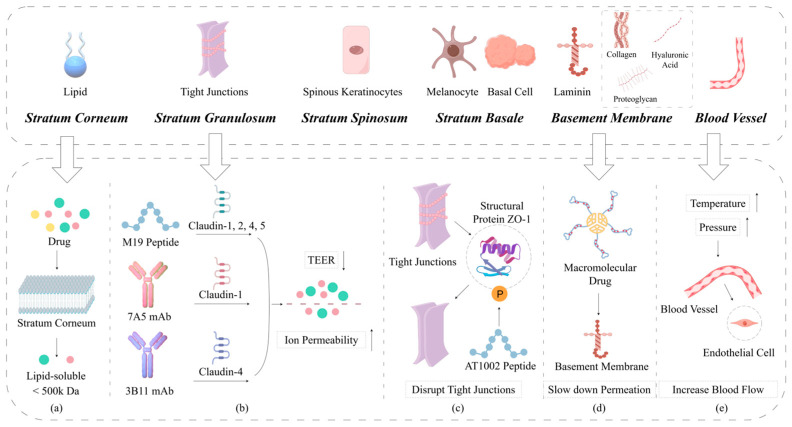
This diagram depicts the obstacles that drugs pass through from the outer layer of the skin to the inner layer, as well as the mechanisms that slow down these obstacles: (**a**) The stratum corneum impedes the passive penetration of large molecules. (**b**) The tight connection of the granular layer is the second barrier for molecules to penetrate beyond the stratum corneum. The binding of certain proteins and peptides can enhance the permeability of tight junctions; M19 peptide binds to claudins (1, 2, 4, and 5); 7A5 monoclonal antibody binds to claudin-1; and 3B11 monoclonal antibody binds to cloudin-4. (**c**) AT1002 peptide phosphorylation tightly linked structural protein ZO-1 enhances drug transdermal permeability. (**d**) The basement membrane slows down the diffusion of larger molecules. (**e**) Endothelial cells are the barrier between blood flow and surrounding skin tissue. Pressure or elevated body temperature leads to increased blood flow, and drugs are more effectively absorbed into the circulatory system.

**Figure 2 pharmaceutics-17-00746-f002:**
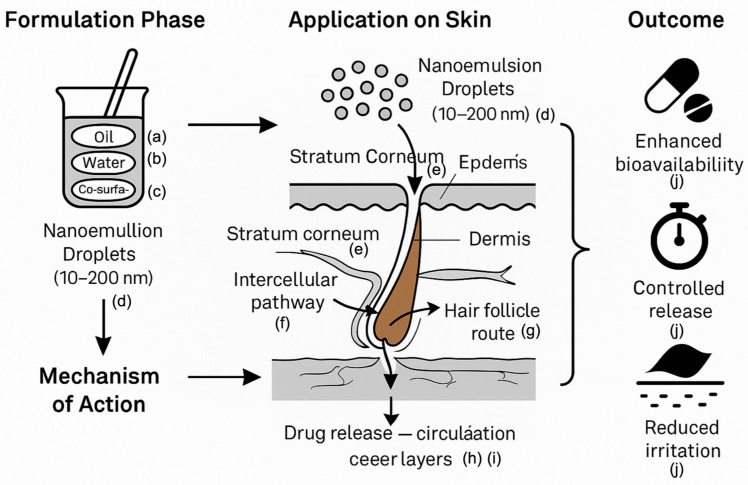
This illustration mechanisms of a nanoemulsion (NE) system: (a) oil phase—solubilizes lipophilic drugs; (b) aqueous phase—carries hydrophilic drugs; (c) surfactant/co-surfactant—stabilizes nanoemulsion droplets; (d) nanoemulsion droplets—spherical structures (10–200 nm) carrying the drug payload; (e) stratum corneum—outermost skin layer; primary barrier to drug permeation; (f) intercellular lipid pathway—main route of diffusion between keratinocytes; (g) hair follicle route—bypasses corneum barrier via follicular openings; (h) drug release zone—area where the active pharmaceutical ingredient (API) is deposited in viable epidermis or dermis; (i) blood capillaries—allow systemic uptake for drugs intended for circulation; (j) therapeutic outcomes—includes enhanced bioavailability, controlled release, and reduced irritation.

**Table 1 pharmaceutics-17-00746-t001:** Recent advances in TDDS for the treatment of ISD.

Study	Transdermal Technique	Type of ISD	Drug	Outcome	Reference
Rana Obaidat et al., 2022	Nanofiber (NF) microfiber system	Different skin inflammatory conditions	Pioglitazone	The flux of PGZ was enhanced by five times using NFs compared to the casted film. PGZ is highly retained in the skin layers, which could be beneficial for achieving local delivery of PGZ into the skin for the management of skin conditions, with minimal amount reaching the blood circulation. A change in the diameter and re-crystallization of PGZ occurred upon storage.	[145]
Ji Heun Jeong et al., 2019	A pullulan hydrogel incorporating Rhus verniciflua extract	Atopic dermatitis	Rhus verniciflflua extract	RVE@PH exerts therapeutic effects through dual functions: the hydrogel film-mediated physical and the RVE-mediated pharmaceutical actions.	[146]
Wenyi Wang et al., 2016	P407/CMCS composite thermosensitive hydrogel	Atopic dermatitis	P407/CMCS hydrogel	FE-SEM images showed that CMCs can endow the hydrogel formulation of P407/CMCs with excellent porous structures, which was found to facilitate the diffusional release of CM across the skin. The sol-gel transition temperatures of P407/CMCs composite hydrogel can be tailored simply by controlling the addition of CMCs without causing the alteration of the sol-gel transition property. The rheological property, however, showed significant change, with a remarked increase in the storage modulus and complex viscosity.	[147]
Yuan Yang et al., 2022	Conductive transdermal drug delivery system	Atopic dermatitis-related skin inflammation; enables on-demand drug loading and release via electrical stimulation	Various drug types integrated with biosensors in smart drug delivery systems	Achieve on-demand release drug by ES and regulate the drug release rate by applying different potentials; c-TDDS can be used as a versatile device with a high drug-loading capacity and electrotriggered drug release profile to effectively deliver one or more anionic drugs, cationic drugs, and neutral drugs for the treatment of many diseases, such as arthritis, mental illness, and analgesia.	[148]
Jeong Hae Choi et al., 2017	Low-temperature atmospheric-pressure plasma-Jaun ointment	Atopic dermatitis	Jaun ointment (JO)	Enhance the drug penetration; regulate the activity of NFκB.	[149]
Meng Yang et al., 2017	Triptolide nano-lotion gel	Dermatitis	Triptolide	Replenish keratin; alter the stratum corneum structure by disrupting it; modify lipid arrangements; moisturize keratin; enhance skin drug absorption; and mitigate adverse digestive, urinary, and reproductive system reactions associated with oral triptolide administration. The TPL-nanoemulsion gels provided higher percutaneous amounts than other carriers did.	[150]
Muhammad Shahid Latif et al., 2022	Transdermal patches made of CMC-Na and HPMC	Psoriasis-induced skin inflammation	Methotrexate	CMC-Na at a high concentration mainly affected proteins (ceramides and proteins) of the skin, resulting in higher penetration and retention of methotrexate in the skin. Reduced serum concentrations; led to better patient compliance; and reduced systemic toxicities.	[151]
Muhammad Shahid Latif et al., 2021	Methotrexate-containing patch using HPMC and ethyl cellulose	Psoriasis	Methotrexate	Among all formulated patches (F1–F9), the F5 formulation exhibited the best in vitro drug release pattern and ex vivo drug permeation ability, having the highest deposition of methotrexate compared to other formulated patches.	[152]
Huaiji Wang et al., 2021	Hyaluronic acid (HA)-based microneedle patch (HM/MN patch) loaded with methotrexate nanoparticles	Psoriasis	Methotrexate nanoparticles	Reduce skin thickening in psoriasis, lower IL-6 and TNF-α levels in the body, mitigate inflammation more effectively than traditional methods, and target the immune system with fewer side effects.	[153]
Yongjian Song et al., 2024	Thermoresponsive hydrogel derived from ionic liquid microemulsion (IL microemulsion)	Psoriasis	Methotrexate	Enhance skin affinity, strengthen formulation adhesion, improve bioavailability, target drug delivery, and extend release characteristics.	[154]
Eman Zmaily Dahmash et al., 2024	Thymoquinone polyamide-based arginine (TQ-Arg-PA) nanocapsules	Psoriasis	Thymoquinone	The TQ-Arg-PA nanocapsules were incorporated into transdermal patches containing EVA, Eudragit E100, glycerin, Span 60, and aloe vera as a penetration enhancer. The patches containing aloe vera provided good penetration enhancement, with an increase in thymoquinone fux (Jss) of 42.64%.	[155]
Wang, W., et al., 2023	Nano-transdermal delivery system combining 9-ol and TCeO_2_ nanoparticles	Psoriasis	TCeO2-TRA-FNL	Integrate TRA to prevent excessive keratinocyte growth and TCeO_2_ to target mitochondria and reduce reactive oxygen species (ROS) levels and inflammation.	[156]
Zi-Ying Zhan et al., 2024	Parthenolide with transdermal administration	Psoriasis	Parthenolide	Par down-regulated the expression of IL-36 and improved psoriasis-like skin inflammation induced by imiquimod.	[157]
Yulin Hua et al., 2022	Methoxypol-yethylene glycol thioether thiol-calcipotriol (mPEG-SS-CPT, PSC) nanomicelle TDDS en-capsulates calcipotriol	Psoriasis	Calcipotriol	ROS sensitivity, good biocompatibility, safe route of administration, and short treatment cycle.	[158]
Hyun Jeong Ju et al., 2025	Hyaluronic acid-based dissolving microneedle patches	Psoriasis	Hyaluronic acid	Sufficient skin penetration strength and enhanced drug delivery in the in vitro study.	[160]
Chaoxiong Wu et al., 2025	Compensatory effect-based oxidative stress management microneedle	Psoriasis	DNA nanostructures	Manage ROS levels, inhibit pyroptosis and abnormal immune activation, modulate ROS levels, and enhance the therapeutic impact of IL-17A siRNA.	[161]
Laurent L’homme et al., 2017	Atromentin-1-derived peptide protein tyrosine phosphatase (AP-PTPP) conjugate	Psoriatic dermatitis; contact dermatitis animal	The astrotactin 1-derived peptide (AP) to EGFP and dTomato	TC-PTP dephosphorylates a wide range of proteins, such as various JAKs and STATs8. Conjugation of a truncated TC-PTP, containing only the protein tyrosine phosphatase domain, with AP (AP-rPTP) led to a functional cell-penetrating phosphatase. AP-rPTP decreased the phosphorylation of STATs induced by cytokines in keratinocytes in vitro. Skin thickening and inflammation in conditions like contact dermatitis and psoriatic dermatitis were alleviated.	[159]
Sopan Nangare et al., 2021	Transfer gel formulation of mulberry leaf extract	Acne	Novel freeze-dried mulberry leaf extract	The optimized batch MF5 provided 86.23% entrapment efficiency of quercetin in the vesicles and 95.79% drug release. The MG1 formulation provided superior antioxidant activity, drug content, and entrapment efficiency, ex vivo drug release, spreadability, homogeneity, and stability to MG2. quercetin in the extract and gel formulation was confirmed by using high-performance thin-layer chromatography. Skin permeability and antioxidant activity were improved.	[164]
Qi Wang et al., 2024	High-molecular-weight hyaluronic acid (HA)-based mi-croneedle system enriched with eugenol	Acne	Eugenol	A transdermal delivery function of cavity-loaded eugenol achieved by rapid dissolution after insertion into the skin tissue. Used for favorable photothermal properties, blood compatibility, cytocompatibility, in vivo biocompatibility, the promotion of cell proliferation, and migration of fibroblasts. Delivers eugenol, exhibits photothermal properties, has antibacterial activity against acne-causing bacteria, promotes sebaceous gland atrophy, reduces inflammation, and aids skin healing.	[165]

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
