# Peer review of "Advances in Transdermal Drug Delivery Systems and Clinical Applications in Inflammatory Skin Diseases"

_pharmaceutics, 2025, doi:10.3390/pharmaceutics17060746_

Round 1

Reviewer 1 Report

Comments and Suggestions for Authors

The manuscript explores the use of transdermal drug delivery systems (TDDS) in the treatment of inflammatory skin diseases—a subject of considerable scientific and clinical relevance. Overall, the manuscript is written in a satisfactory manner. The abstract effectively underscores the limitations of conventional therapeutic approaches and positions TDDS as a compelling alternative due to their capacity for localized, sustained, and minimally invasive drug administration. The stated aim to review clinical data, mechanisms of action, and translational potential is both valuable and timely. However, the current version of the manuscript falls short in fully realizing these goals. Several critical areas require improvement to enhance its scientific rigor and overall clarity.

Specific Comments and Recommendations

1. Title Alignment and Use of Visual Aids:

The title of the manuscript does not fully align with the depth and scope of content presented. Consider revising the title to better reflect the manuscript's focus. Additionally, the inclusion of relevant figures and summary tables would significantly improve the readability and utility of the manuscript, allowing for easier comparison of TDDS technologies and trends in their development.

2. Mismatch Between Abstract and Content:

While the abstract suggests a comprehensive review encompassing clinical data, real-world application, and future directions, the main body lacks this level of depth. The discussion remains general and does not critically evaluate or synthesize specific clinical findings or mechanistic studies.

3-Missing Figure Caption (Page 4):

A caption for the figure on page 4 is missing. Please provide a detailed and descriptive caption to enhance clarity and context.

4-Inadequate Referencing in Introduction:

The introduction would benefit from the inclusion of appropriate references, particularly when describing the structure of the skin and its relevance to TDDS.

5-Missing or Incomplete References:

Several parts of the manuscript lack citations where claims are made. Proper referencing is essential to guide the reader and support scientific assertions. Please ensure the following lines include appropriate citations, such as

Page 5, lines 185 and 189

Page 5, lines 213–214 and line 218

Page 7, line 328

Page 8, lines 348–349 and 364

Page 12, line 550

Page 13, line 599

Page 15, line 708

Clarification of Microneedle Terminology:

On page 8 (lines 348–349 and 364), the claim that microneedle application is "painless" should be reconsidered, as this may not be universally accurate. Furthermore, in lines 363–367, the terms microneedling and microneedle patches are used interchangeably, which is misleading. These are distinct techniques with different applications and mechanisms of action. Please revise accordingly.

Author Response

Dear reviewer,

Thank you very much for your comments and professional advice. These opinions help to improve academic rigor of our article. Based on yoursuggestion and request, we have made corrected modifications on the revised manuscript in red. We hope that our work can be improved again.

Title Alignment and Use of Visual Aids: 

I have changed the title of the article from “Application and progress of transdermal drug delivery system in inflammatory skin diseases”to“Advances in Transdermal Drug Delivery Systems and Clinical Applications in inflammatory skin diseases”to reflect the content of the article more accurately.

  1. Mismatch Between Abstract and Content:

The following is my revised abstract:

Inflammatory skin diseases are highly prevalent conditions characterized by complex immune responses that result in skin tissue damage and pain, significantly impacting patients' physical health.Traditional therapeutic approaches, including oral administration and injections, continue to exhibit inherent limitations.Consequently, there is growing interest in exploring alternative drug delivery systems that offer more effective, targeted, and patient-friendly therapeutic options. Transdermal administration emerges as a promising solution for managing inflammatory skin diseases, facilitating sustained drug release, reducing the frequency of dosing.This review provides a comprehensive overview of the skin barrier and critically summarizes clinically adopted transdermal drug delivery systems (TDDS), including sonophoresis, iontophoresis, chemical penetration enhancers, and electroporation. Particular emphasis is placed on emerging advances in microneedle- and nanocarrier-facilitated transdermal delivery strategies. Moreover, the article synthesizes recent clinical evidence regarding the application of TDDS in the treatment of atopic dermatitis, psoriasis, and acne. This review examines clinical studies evaluating various transdermal drug delivery systems for the treatment of major inflammatory skin diseases, with an emphasis on their mechanisms of action, advantages, challenges, and future directions. Transdermal drug delivery systems hold the potential to deliver more efficient and safer treatment and management strategies for patients afflicted with inflammatory skin diseases.

3-Missing Figure Caption (Page 4):

I have added Figure Caption as followed:

This diagram depicts the obstacles that drugs pass through from the outer layer of the. skin to the inner layer, as well as the mechanisms that slow down these obstacles: (a) The stratum corneum impedes the passive penetration of large molecules; (b) The tight connection of the granular layer is the second barrier for molecules to penetrate beyond the stratum corneum. The binding of certain proteins and peptides can enhance the permeability of tight junctions; M19 peptide binds to claudins (1, 2, 4, and 5) ; 7A5 monoclonal antibody binds to claudin-1; 3B11 monoclonal antibody binds to cloudin-4; (c) AT1002 peptide phosphorylation tightly linked structural protein ZO-1 enhances drug transdermal permeability; (d) The basement membrane slows down the diffusion of larger molecules; (e) Endothelial cells are the barrier between blood flow and surrounding skin tissue. Pressure or elevated body temperature leads to increased blood flow, and drugs are more effectively absorbed into the circulatory system.

Finally, I have added new references to provide strong evidence for the viewpoints presented in the paper, highlighted them in red in the main text, and revised the inappropriate statements you raised.

Thank you very much for your attention and time.Looking forward to hearing

from you.

                                                         Yours sincerely,

Sizhuo Liu1, Tinghan Deng2, Hongbin Cheng3, Jun Lu4,* and Jingping Wu5,*

                                                            27/04/2025

Reviewer 2 Report

Comments and Suggestions for Authors

The use of abbreviations throughout the text should be standardized, as "TDDS" is mentioned multiple times in its abbreviated and full form. Please polish the text.

Additionally, the study would benefit from including tables and figures, facilitating a more efficient and engaging reading experience. A comparative table summarizing key characteristics of the selected studies—such as formulation type, population, clinical condition, treatment modalities, and main outcomes—would enhance clarity.

The figure presented on page 4 is missing a descriptive caption that outlines its key attributes and contributions. It is imperative that this figure be referenced within the text and that the layer strata be italicized both in the figure and in the written content. The specific software utilized to create the figure should also be included in the caption.

Furthermore, the absence of a clearly defined search strategy detracts from the study’s transparency. A detailed explanation of the eligibility criteria employed by the authors is essential for reproducibility and understanding the selection process.

The central research question that underpins this review needs to be clearly articulated and addressed within the text. 

While the review presents interesting insights, it suffers from excessive verbosity. Although thorough explanations are necessary, the authors should strive to eliminate superfluous details and repetitions to improve readability and coherence. The current iteration's dense presentation could confuse readers.

Author Response

Response to Dear Reviewer :

We are grateful for your insightful and constructive comments. In response, we have carefully revised the manuscript to address each point you raised. Please find our detailed point-by-point responses below:

We have standardized the use of the term “Transdermal Drug Delivery System (TDDS)” throughout the manuscript. The full term is defined upon first mention, and only the abbreviation “TDDS” is used thereafter. Additionally, we have polished the entire text to improve clarity, consistency, and academic tone. 

We have added a new table summarizing the key characteristics of selected studies, including formulation types, target populations, clinical indications, treatment strategies, and major outcomes. This table is intended to improve the structure and accessibility of the review. It has been included in Section 4.3 and is titled “Recent Advances in TDDS for the Treatment of ISD” (pages 15–25).

We have revised the figure on page 4 as follows:

A descriptive caption summarizing its key components and purpose has been added.

Stratigraphic layers have been italicized in both the figure and the relevant text sections.

The figure is now properly referenced in the main text.

The caption also includes the name of the software used to create the figure (Figdraw).

All these revisions have been marked in red for ease of review.

We have revised the Abstract and Introduction sections to clearly articulate the central research question guiding this review. Specifically, we now emphasize the need to explore recent progress in TDDS for treating inflammatory skin diseases (ISD), with a focus on the mechanisms, formulation strategies, and clinical translation potential. These revisions are intended to provide the manuscript with a more focused narrative structure.

We have carefully revised the manuscript to streamline the text, remove redundant information, and improve readability. Particular attention was paid to eliminating repetitive statements and condensing overly detailed explanations without sacrificing key content. These edits were made throughout the manuscript and are highlighted in red for your review.

Thank you very much for your attention and time.Looking forward to hearing

from you.

                                                                                                               Yours sincerely,

Sizhuo Liu1, Tinghan Deng2, Hongbin Cheng3, Jun Lu4,* and Jingping Wu5,*

                                                                                                                      06/05/2025

Round 2

Reviewer 1 Report

Comments and Suggestions for Authors

A clear distinction must be made between microneedling and microneedle patches (e.g., dissolvable, solid, hollow), as they are substantially different techniques and should not be used interchangeably. Section 3.5 requires revision or a complete rewrite to accurately reflect this distinction.

Additionally, typographical errors are present in lines 505 and 861 and should be corrected.

Author Response

We sincerely appreciate the reviewer’s insightful comments and constructive suggestions.

Comment 1: A clear distinction must be made between microneedling and microneedle patches (e.g., dissolvable, solid, hollow), as they are substantially different techniques and should not be used interchangeably. Section 3.5 requires revision or a complete rewrite to accurately reflect this distinction.

Response 1:Thank you for this valuable observation. In response, we have thoroughly revised Section 3.5 to clearly differentiate between microneedling and microneedle patches. The revised section now defines microneedling as a minimally invasive dermatological procedure that enhances skin permeability through the creation of microchannels, whereas microneedle patches are described as self-contained transdermal drug delivery systems capable of delivering therapeutic agents directly through the skin. We have elaborated on the functional classifications of microneedle patches (e.g., solid, coated, dissolving, hollow, and hydrogel-forming) and emphasized their mechanistic differences and distinct clinical applications.We believe this revised version better reflects the scientific nuances between the two technologies and addresses the reviewer’s concern.

Comment 2: Typographical errors are present in lines 505 and 861 and should be corrected.  

Response 2: Thank you for pointing this out. We have carefully reviewed the manuscript and corrected the typographical errors. A thorough proofreading has also been conducted to ensure consistency and clarity throughout the text.   All changes have been marked in red in the revised manuscript for your convenience.

Reviewer 2 Report

Comments and Suggestions for Authors

This round of revisions consistently improved the quality of the document. However, there are still two major points that requires attention:

1) Please ensure that you provide the source of any images or the software used to create any original figures.

2) It is crucial to explain in the manuscript how you selected the articles for this review. Without this information, it becomes challenging to understand how you ensured that the relevant and current literature was adequately covered.

Author Response

Response to Reviewer   We are grateful for the reviewer’s positive feedback and for recognizing the improvements made in this revision. We also appreciate the reviewer’s continued efforts to enhance the quality and transparency of our manuscript. In response to the two major points raised:   Comment 1: Please ensure that you provide the source of any images or the software used to create any original figures.   Response 1: Thank you for this important reminder. We have now included a clear statement in the figure captions specifying that the original schematic illustrations were created by the authors using Figdraw (https://www.figdraw.com), a licensed online tool designed for scientific figure generation. These clarifications have been added to all relevant figure legends.   Comment 2: It is crucial to explain in the manuscript how you selected the articles for this review. Without this information, it becomes challenging to understand how you ensured that the relevant and current literature was adequately covered.   Response 2: We fully agree with the reviewer’s point. To address this, we have added a detailed description of our article selection strategy to the revised manuscript. Specifically, we conducted an advanced search on PubMed using the search string: (Transdermal Drug Delivery System[Title/Abstract]) AND ((Psoriasis[Title/Abstract]) OR (Acne[Title/Abstract]) OR (Atopic Dermatitis[Title/Abstract])), with the latest update performed in May 2025. The inclusion criteria and total number of retrieved articles have been stated explicitly in the revised text. This addition enhances the transparency and reproducibility of our literature selection process.   All changes have been highlighted in red in the revised manuscript for easy reference.
